# Spatial transcriptomics reveal topological immune landscapes of Asian head and neck angiosarcoma

Jui Wan Loh[1], Jing Yi Lee[1], Abner Herbert Lim [2], Peiyong Guan[3], Boon Yee Lim[1], Bavani Kannan [1], Elizabeth Chun Yong Lee[1], Ning Xin Gu [4], Tun Kiat Ko[1], Cedric Chuan-Young Ng [1], Jeffrey Chun Tatt Lim[5], Joe Yeong[5], Jing Quan Lim [6], Choon Kiat Ong[6,7], Bin Tean Teh [2,5,7✉] & Jason Yongsheng Chan [1,8,9✉]

Angiosarcomas are rare malignant tumors of the endothelium, arising commonly from the head and neck region (AS-HN) and recently associated with ultraviolet (UV) exposure and human herpesvirus-7 infection. We examined 81 cases of angiosarcomas, including 47 cases of AS-HN, integrating information from whole genome sequencing, gene expression profiling and spatial transcriptomics (10X Visium). In the AS-HN cohort, we observed recurrent somatic mutations in *CSMD3* (18%), *LRP1B* (18%), *MUC16* (18%), *POT1* (16%) and *TP53* (16%). UV-positive AS-HN harbored significantly higher tumor mutation burden than UV-negative cases ($p = 0.0294$). NanoString profiling identified three clusters with distinct tumor inflammation signature scores ($p < 0.001$). Spatial transcriptomics revealed topological profiles of the tumor microenvironment, identifying dominant but tumor-excluded inflammatory signals in immune-hot cases and immune foci even in otherwise immune-cold cases. In conclusion, spatial transcriptomics reveal the tumor immune landscape of angiosarcoma, and in combination with multi-omic information, may improve implementation of treatment strategies.

[1] Cancer Discovery Hub, National Cancer Centre Singapore, Singapore, Singapore. [2] Laboratory of Cancer Epigenome, Division of Medical Sciences, National Cancer Centre Singapore, Singapore, Singapore. [3] Genome Institute of Singapore, Agency of Science, Technology and Research (ASTAR), Singapore, Singapore. [4] MGI Tech Singapore PTE LTD, Singapore, Singapore. [5] Institute of Molecular and Cell Biology, Agency of Science, Technology and Research (ASTAR), Singapore, Singapore. [6] Lymphoma Genomic Translational Research Laboratory, Division of Cellular and Molecular Research, National Cancer Centre Singapore, Singapore, Singapore. [7] Program in Cancer and Stem Cell Biology, Duke-NUS Medical School, Singapore, Singapore. [8] Division of Medical Oncology, National Cancer Centre Singapore, Singapore, Singapore. [9] Oncology Academic Clinical Program, Duke-NUS Medical School, Singapore, Singapore. ✉email: teh.bin.tean@singhealth.com.sg; jason.chan.y.s@nccs.com.sg

Angiosarcomas belong to a rare but aggressive subgroup of malignant soft tissue tumors derived from endothelial cells. Although rare, angiosarcomas have been associated with several risk factors, such as exposure to ionizing radiation, ultraviolet (UV) radiation or exogenous toxic chemicals, as well as the presence of chronic lymphedema or familial syndromes[1,2]. Clinically, they can develop in different anatomical structures, including the head and neck (AS-HN), breast, viscera, trunk, and extremities. Recent studies suggest that angiosarcomas harbor a unique geographic distribution, demonstrating higher prevalence in Asian countries compared to the West[3,4]. Furthermore, AS-HN appears to be more common amongst Asian populations and may be etiologically related to UV mutagenesis, as well as the human herpesvirus-7 (HHV-7)[5,6].

Contemporary treatment of advanced angiosarcomas, including AS-HN, remains highly challenging with limited options, often leading to poor outcomes. The use of chemotherapeutic agents such as anthracyclines and taxanes, as well as small molecule multi-kinase inhibitors are characteristically met with rapid resistance and treatment failure[6,7]. Immune checkpoint inhibitors targeting PD-1/PD-L1 or CTLA-4 have recently shown promising clinical efficacy in angiosarcoma therapy. In the multicentre phase II trial conducted by SWOG, dual immune checkpoint blockade with ipilimumab (anti-CTLA-4) and nivolumab (anti-PD-1) reported objective responses in four of 16 patients (25%). Notably, three out of five patients (60%) with primary cutaneous scalp or face tumors achieved a confirmed response[8]. These findings mirror observations from retrospective cohort studies reporting promising activity of pembrolizumab (anti-PD-1)[9,10]. Potential biomarkers such as high tumor mutation burden (TMB), high PD-L1 protein expression on immunohistochemistry, and an immunogenic tumor microenvironment have been proposed, yet does not robustly predict treatment response[9,11,12]. The genomic and immune landscape of angiosarcomas thus require deeper characterization in order to better explain the potential reasons for disparate responses to checkpoint immunotherapy.

In this study, we investigated the genomic, transcriptomic and immune landscapes of angiosarcomas from an Asian population, focusing on AS-HN. Spatial transcriptomic profiling using the 10X Genomics Visium platform was performed on representative cases characterized by presence or absence of UV signatures and/or HHV-7, revealing the topological tumor immune landscape of angiosarcoma.

## Results

### Somatic mutations and copy number alterations in head and neck angiosarcoma

We performed whole genome sequencing (WGS) of 76 angiosarcomas, including 44 from the head and neck region and 32 from other anatomical origins (Supplementary Fig. 1 and Supplementary Data 1). In the AS-HN cohort, we observed recurrent somatic mutations in *CSMD3* (18%), *LRP1B* (18%), *MUC16* (18%), *POT1* (16%) and *TP53* (16%) (Fig. 1a and Supplementary Data 2). The median TMB was 4.28 mutations per coding megabase (mt/Mb) (range, 0.03–13.4). UV and HHV-7 status were assessed by immunohistochemistry (IHC) as previously reported[5]. TMB was significantly higher in UV compared to non-UV-related AS-HN (median, 4.63 vs 3.30 mt/Mb, $p = 0.0294$). No significant difference in TMB was observed between HHV-7-positive and HHV-7-negative tumors (Fig. 1b). The estimated proportions of mutations in each of six possible base substitution classes, as well as the proportions of mutations contributed by each inferred mutational signature in individual samples were examined. The COSMIC Mutational Signature for exposure to UV radiation, as characterized by a majority of C:T>T:C single-base substitutions (SBS), was observed in 23 cases (SBS 7a and/or 7b, 52.3%) (Fig. 2).

UV-positive cases as assessed by IHC harbored a higher proportion of SBS 7 signatures (median, 20%; range, 0–96%) compared to UV-negative cases (median, 0%; range, 0–78%), though this did not meet statistical significance ($p = 0.0671$). Other mutational patterns identified included SBS 1, SBS5, SBS38, and SBS 40 (Fig. 2).

### Three distinct clusters of AS-HN defined by immune microenvironmental and tumor-related pathways

The 770-gene NanoString PanCancer IO360 panel was used to profile a set of 67 angiosarcomas, identifying clusters defined by immune microenvironmental and tumor-related pathways. The immune-hot cluster was characterized by relative upregulation of several immune-related pathways. The immune-intermediate cluster exhibited upregulation of several oncogenic signaling pathways, particular in intermediate-B over intermediate-A. The immune-cold cluster demonstrated a relatively bland immune-oncogenic gene expression profile. As inferred using NanoString gene expression data, AS-HN were relatively enriched for mast cells (Supplementary Fig. 2 and Supplementary Data 3 and 4).

Amongst the AS-HN subgroup ($n = 40$), three clusters with distinct levels of immune-related signaling were again identified (Fig. 3a). The Tumor Inflammation Signature (TIS) scores, which have been suggested as a potential pan-cancer predictive biomarker of checkpoint immunotherapy response[13], were highest in the immune-hot cluster, as compared with immune-intermediate and immune-cold (7.82 vs. 7.49 vs. 5.90, respectively; Kruskal–Wallis, $p = 0.00021$) (Fig. 3b). We examined these samples and compared with available data on immune cell infiltration from our prior publication[5], obtained using multiplex immunohistochemistry/immunofluorescence (mIHC/IF). We observed that the proportion of immune cells (CD8+ cytotoxic T-cells, CD15+ neutrophils, CD68+ macrophages, FOXP3+ T-reg cells) relative to ERG+ tumor cells directly correlated with the immune clusters ($n = 11$ immune-hot and $n = 5$ immune-cold cases) inferred from NanoString transcriptomic profiling ($p = 0.0201$) (Supplementary Fig. 2). HHV-7-positive tumors were relatively enriched for B-cells and mast cells, and harbored higher TIS scores as compared to HHV-7-negative tumors ($p = 0.0312$) (Fig. 3c, d). UV-positive tumors were relatively enriched for CD8+/cytotoxic T-cells, but had comparable TIS scores to UV-negative tumors (Fig. 3e, f).

### Spatial analysis of head and neck angiosarcoma

To analyze the tumor microenvironmental architecture of AS-HN, we employed the 10X Genomics Visium spatial transcriptomics technology on frozen tissue sections ($n = 4$). This enables the quantification of gene expression across the whole transcriptome at a spatial resolution using spatially-barcoded oligonucleotides located in 55 micron-wide spots. Four samples exhibiting a combination of different UV and HHV-7 status as well as immune signatures, were selected for spatial transcriptomic profiling (AS-17: UV-positive, HHV-7-positive, and immune hot; AS-52: UV-positive, HHV-7-negative, and immune intermediate; AS-03: UV-negative, HHV-7-positive, and immune intermediate; AS-07: UV-negative, HHV-7-negative, and immune cold) (Fig. 4a).

Merged analysis of these four samples detected tumor cells and other microenvironmental elements including fibroblasts, immune cells and keratinocytes (Fig. 4b and Supplementary Fig. 3). Spatial projections of the UMAP spot transcriptome clusters onto their respective tissue sections revealed important information about the tumor microenvironment. Signals from myeloid and other immune cell types were strongly detected in AS-17, the only immune-hot specimen. Collectively, they make up the major composition (86%) of the tissue area, along with fibroblasts and tumor cells. Interestingly, a significant component comprising of predominantly immune cells were observed to be

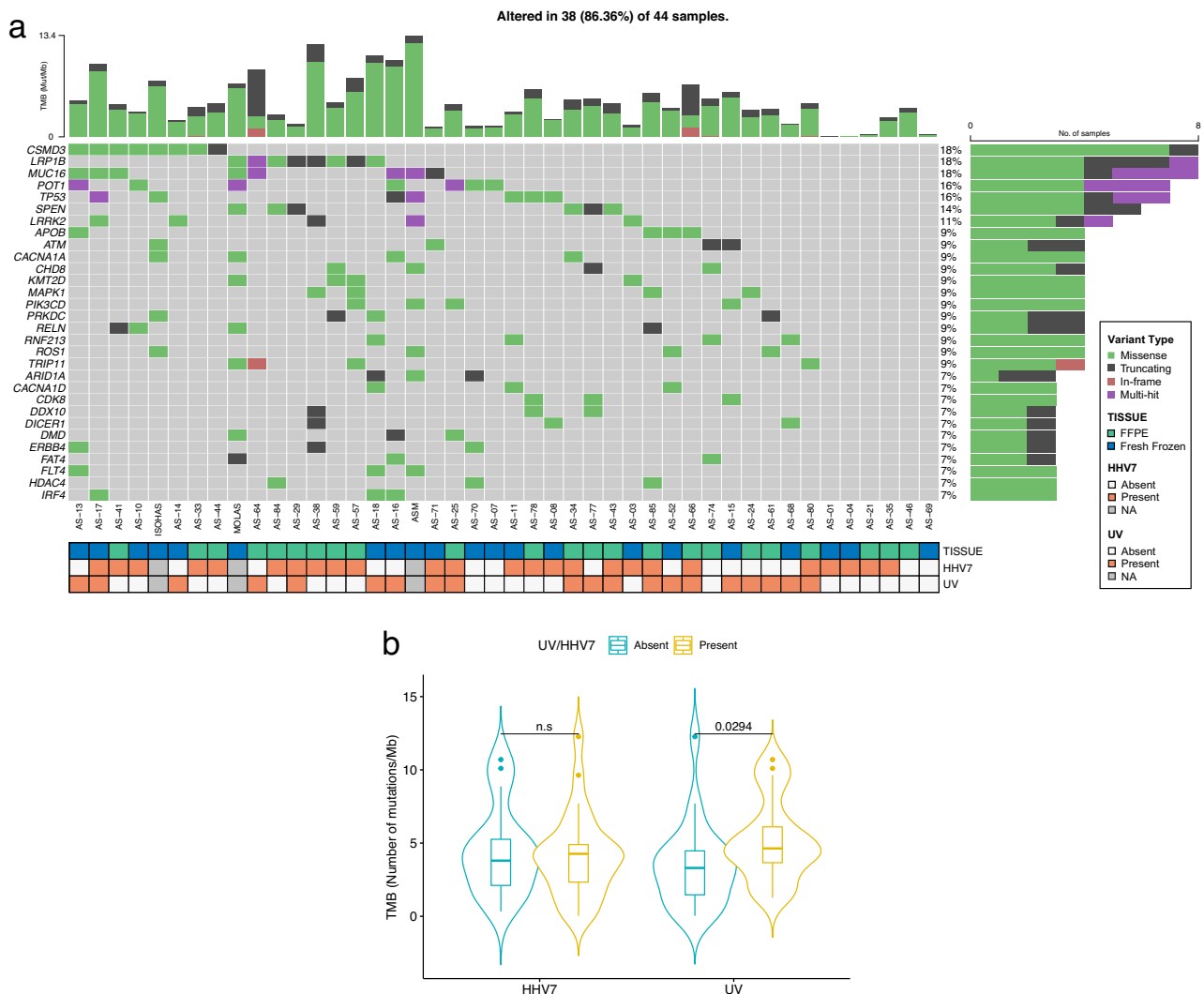

**Fig. 1 Somatic mutational landscape of head and neck angiosarcoma. a** Oncoplot of somatic non-synonymous variants of interest, including recurrent mutations in *CSMD3* (18%), *LRP1B* (18%), *MUC16* (18%), *POT1* (16%), and *TP53* (16%). The median tumor mutation burden (TMB) was 4.28 mutations per coding megabase (mt/Mb) (range, 0.03–13.4). **b** TMB was significantly higher in UV compared to non-UV-related AS-HN (median, 4.63 vs 3.30 mt/Mb, $p = 0.0294$ by Mann–Whitney U test). Violin plots are represented by median and lower to upper quartile values, and the vertical line extends from the minimum to the maximum value. No significant difference in TMB was observed between HHV-7-positive and HHV-7-negative tumors.

largely excluded from the bulk of the tumor compartment, which co-existed instead with a heavy myeloid cell infiltrate (Fig. 4c).

The two immune-intermediate cases, AS-52 and AS-03, were composed mainly of signals derived from tumor cells (>47%) and fibroblasts (>30%) (Fig. 4d, e). AS-03 contained fewer immune-infiltrated fibroblasts than AS-52 (31% vs 56%), and were distributed in a more focal manner in the former and scattered throughout the tumor in the latter. The immune-cold AS-07 comprised predominantly of keratinocytes (68%), in keeping with histological findings. This was followed by immune-infiltrated tumor cells/fibroblasts (28%) (Fig. 4f).

**Spatial distribution of tumor inflammation signatures**. The spatial distribution of tumor, stromal and immune cell populations in each tumor section was inferred based on transcriptomic profiles. The tumor samples were heavily infiltrated with fibroblasts in AS-03 and AS-07, while CD8-positive T-cells, NK cells and macrophages featured prominently in AS-17 (Fig. 5). Corroborating with findings from NanoString gene expression profiling, the relative average expression of the 18 genes informing the TIS score (*PSMB10, CD276, PDCD1LG2, HLA-DQA1, NKG7,*

*HLA-E, TIGIT, CD274, CXCL9, CMKLR1, LAG3, CD26, CXCR6, IDO1, CD8A, HLA-DRB1, CCL5,* and *STAT1*) were highlighted particularly in AS-17, specifically in the immune cell compartment (Fig. 6a). Overall, the relative average TIS expression decreased from AS-17 (0.2290), AS-03 (0.0880), AS-07 (−0.0587), and AS-52 (−0.0580) (Fig. 6b). Immune cells from AS-17 exhibited the highest average expression of TIS genes, followed by immune-filtrated fibroblasts in AS-17, AS-03 and AS-52, as well as immune-infiltrated tumor cells/fibroblasts in AS-07. The myeloid cells had the lowest TIS gene expression in AS-17 (Fig. 6c, d; Supplementary Data 5). The TIS scores could be visualized with respect to their spatial context in each sample. In keeping with earlier analysis, TIS signals coincided with the immune compartment but were largely tumor-excluded in AS-17. The corresponding spots exhibiting higher levels of TIS gene expression exhibited a focal pattern in AS-03 and AS-07, while displaying a scattered pattern in AS-52 (Fig. 6e).

**Discussion**

Angiosarcomas of the head and neck region (AS-HN) represent a distinct anatomical disease subtype characterized by several unique

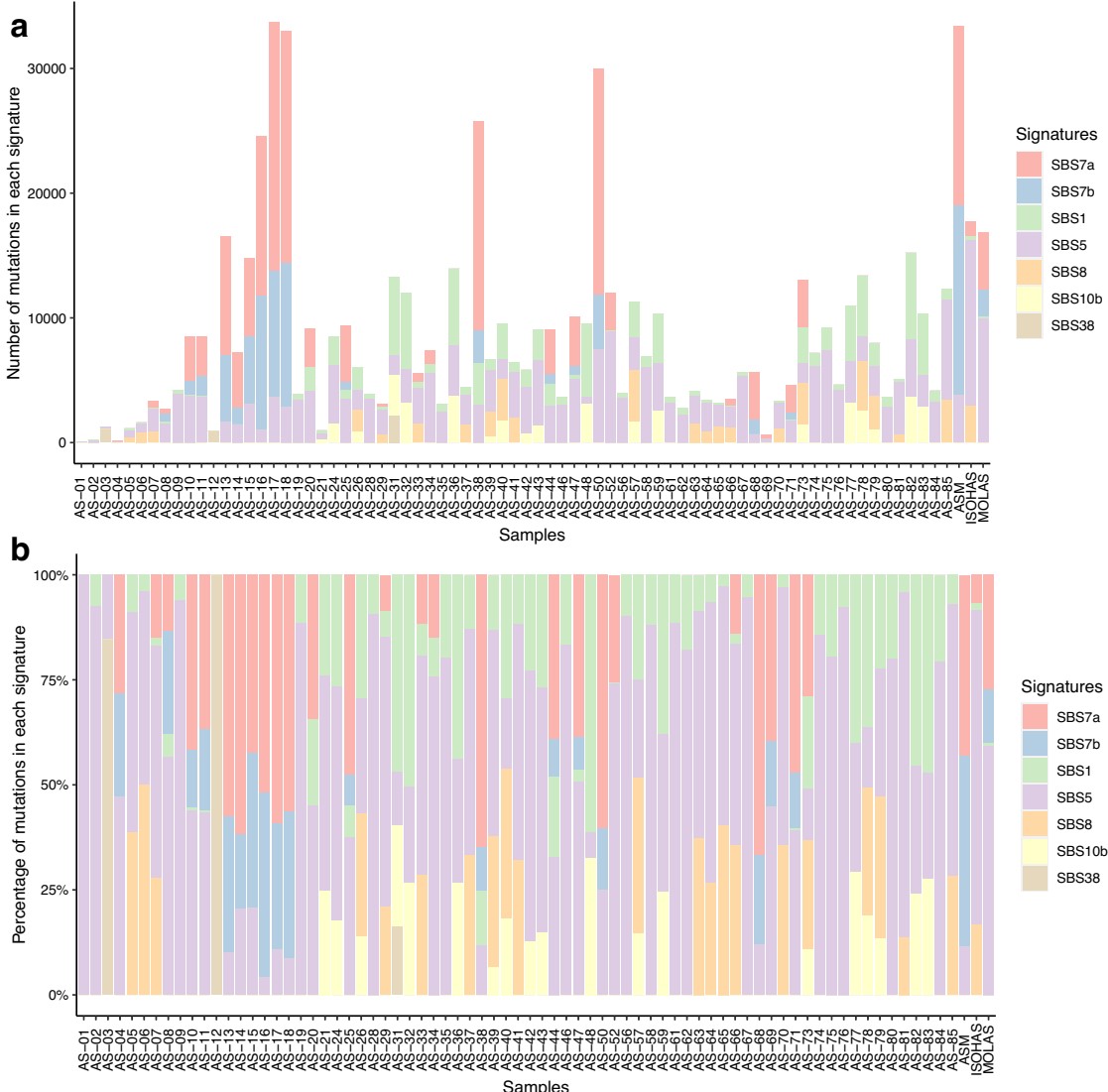

**Fig. 2 Mutational signatures of head and neck angiosarcoma.** AS-HN harbored distinctive UV-related mutational signatures (COSMIC Mutational Signatures 7a and 7b) in 23 cases (SBS 7a and/or 7b, 52.3%). Mutations in each signature per sample are shown by (**a**) absolute number and by (**b**) proportion. Other mutational patterns identified included SBS 1, SBS 5, SBS 38, and SBS 40.

clinicopathological features. Geographically, AS-HN appears to have a predilection in Asian populations, particularly amongst the elderly[2,5,6]. In our recent retrospective study on 150 Asian patients diagnosed with angiosarcomas, most cases (58.7%) originated from the head and neck region. AS-HN were more common in men and amongst elderly above 65 years old[6]. At the genomic level, AS-HN is known to harbor signatures of UV mutagenesis, as well as the presence of HHV-7[2,5]. Previous studies on the genomic landscape of angiosarcoma have described recurrent somatic mutations of angiogenic signaling pathway genes, including *KDR*, *PTPRB*, and *PLCG1*[14–18], as well as genes involved in oncogenic signaling pathways such as *MAPK*, *PIK3CA/AKT/mTOR*, and *TP53*[19,20]. In our current study, we again observed recurrent somatic mutations in *POT1* (16%) and *TP53* (16%) in AS-HN, confirming previous reports[11]. POT1 is a component of the shelterin complex and plays a key role in the regulation of telomere length and chromosomal stability. Missense mutation in *POT1* have been identified in TP53-negative Li-Fraumeni–like families with cardiac angiosarcoma. Mutation carriers demonstrated reduced levels of POT1 bound to telomeres, which correlated with abnormally long telomeres and increased telomere fragility[21,22].

We observed other frequently-mutated genes in AS-HN including *CSMD3*, *MUC16*, and *LRP1B*, which are known cancer-related genes[23,24]. *CSMD3* is a large gene consisting of 73 exons, encoding CUB and Sushi multiple domains 3, a transmembrane protein with multiple CUB and sushi domains. Mutations in *CSMD3* have been reported in dedifferentiated liposarcomas and synovial sarcomas[25,26]. Recently, sarcomas harboring an immune-hot phenotype were demonstrated to carry the highest frequencies of *CSMD3* mutations[27]. *LRP1B* encodes for low-density lipoprotein receptor-related protein 1b, a putative tumor suppressor in which mutations have been described in angiosarcomas[28]. A recent study showed that pathogenic mutations in *LRP1B* were correlated with superior response rates and survival outcomes to immune checkpoint inhibitors across various tumor types, including sarcomas[29].

The use of immune checkpoint inhibitors in the treatment of angiosarcoma has demonstrated promising clinical utility in recent reports, particularly in AS-HN harboring UV mutational signatures and high TMB[2,8]. Notably, in the current study, approximately one-third of AS-HN could be classified as immune-hot tumors harboring high TIS scores, within which a considerable proportion also demonstrated evidence of UV-induced DNA

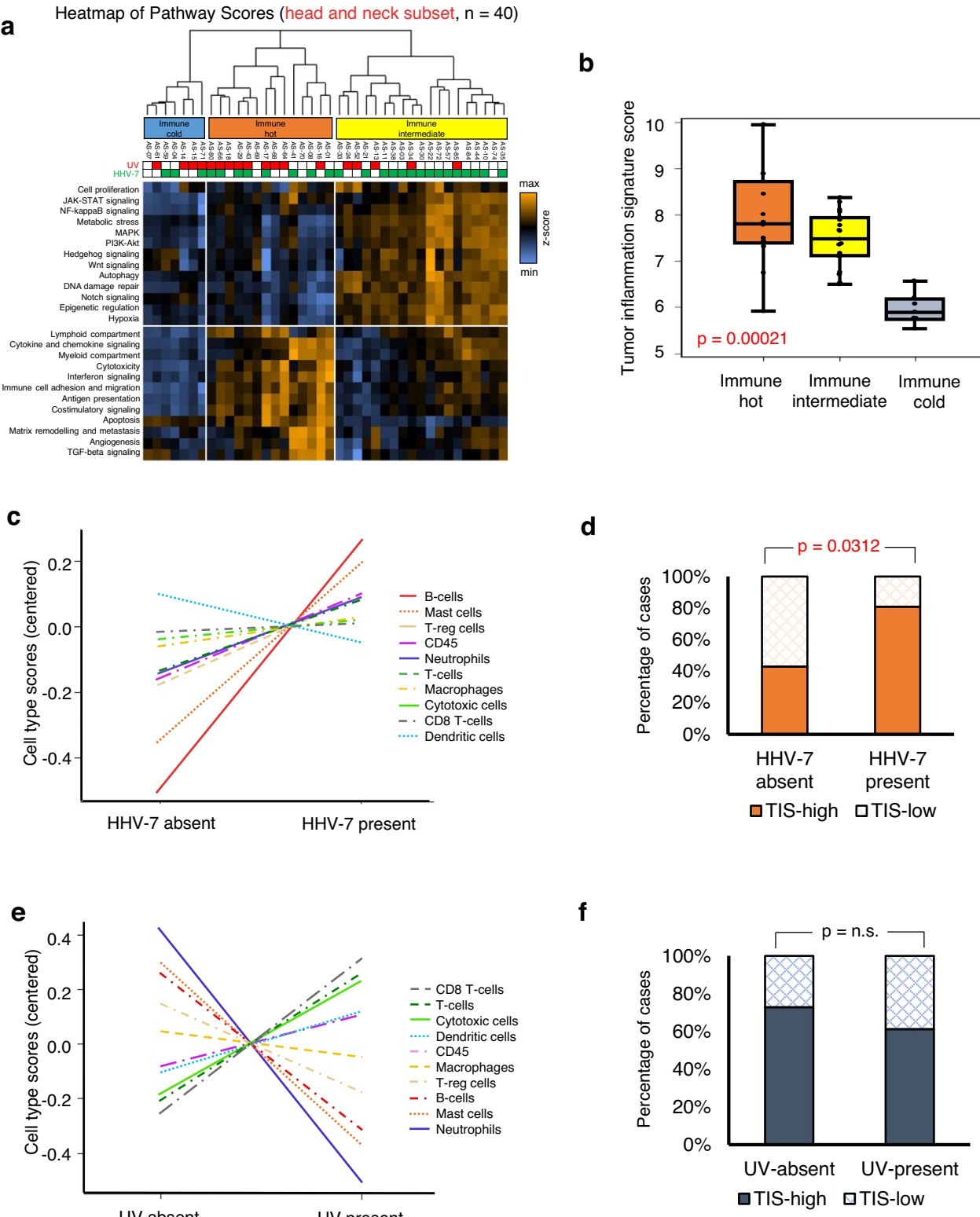

**Fig. 3 Three distinct clusters of head and neck angiosarcomas defined by immune microenvironmental and tumor-related pathways. a** NanoString IO360 panel profiling revealed 3 clusters with distinct levels of immune signaling. The immune-hot cluster was characterized by relative upregulation of several immune-related pathways, while the immune-intermediate cluster exhibited upregulation of several oncogenic signaling pathways. The immune-cold cluster demonstrated a relatively bland immune-oncogenic gene expression profile. Color key: orange indicates high scores; blue indicates low scores. Scores are displayed on the same scale via a Z-transformation. **b** TIS scores were highest in the immune-hot cluster, as compared with immune-intermediate and immune-cold (7.82 vs. 7.49 vs. 5.90, respectively; Kruskal–Wallis, $p = 0.00021$). Box-plots are represented by median and lower to upper quartile values, and the vertical line extends from the minimum to the maximum value. **c** As inferred using NanoString gene expression data, HHV-7-positive tumors were relatively enriched for B-cells and mast cells, and (**d**) harbored higher TIS scores as compared to HHV-7-negative tumors ($p = 0.0312$). **e** UV-positive tumors were relatively enriched for CD8+/cytotoxic T-cells, but (**f**) had comparable TIS scores to UV-negative tumors.

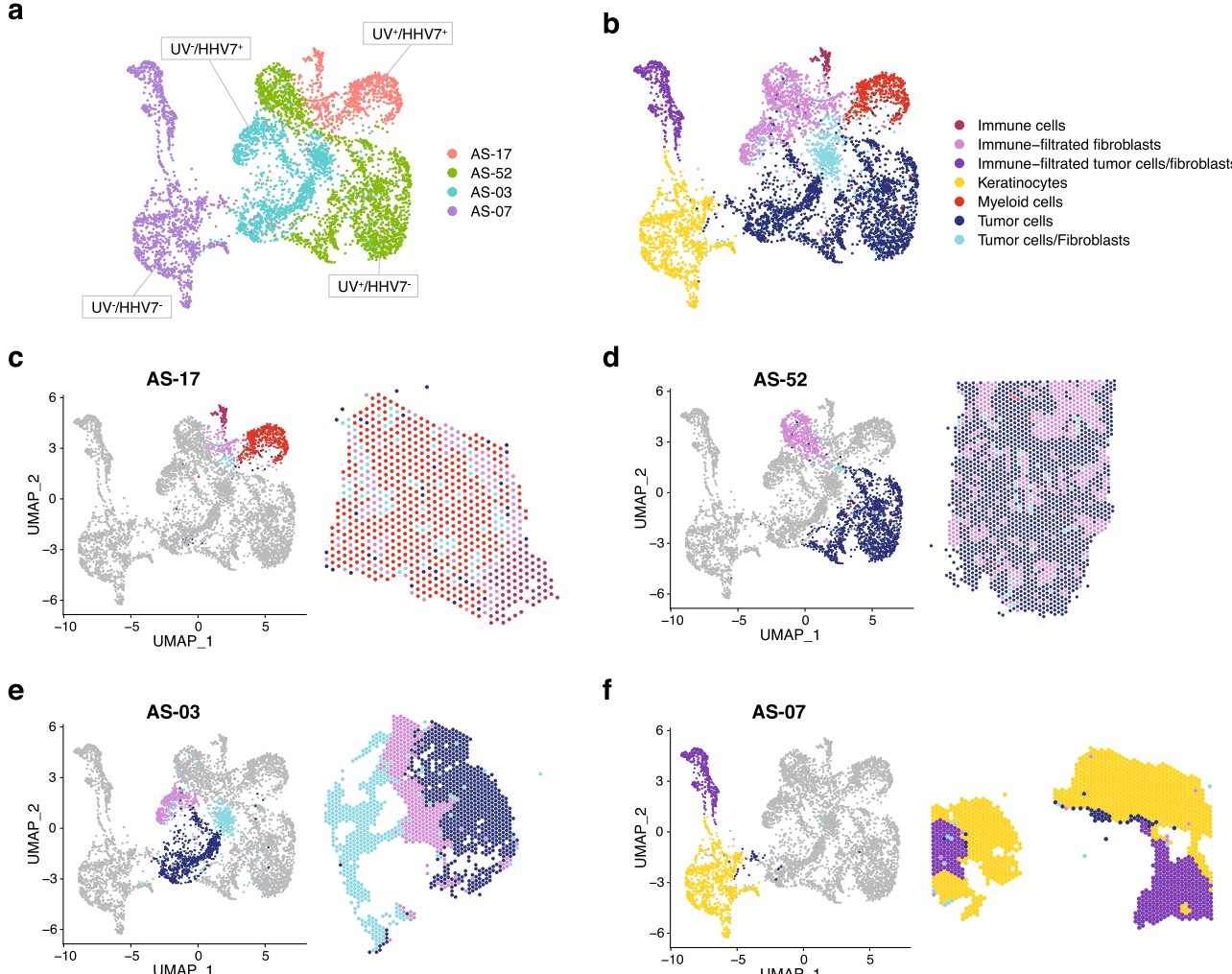

**Fig. 4 Spatial transcriptomic profiling of head and neck angiosarcomas representing four putative pathogenic etiologies.** UMAP plots of merged spatial datasets showing (**a**) the distribution by samples (AS-17: UV-positive, HHV-7-positive, and immune hot; AS-52: UV-positive, HHV-7-negative, and immune intermediate; AS-03: UV-negative, HHV-7-positive, and immune intermediate; AS-07: UV-negative, HHV-7-negative, and immune cold) and (**b**) the distribution of seven clusters annotated with their predominant cell types. **c**–**f** Spatial projections of the UMAP spot transcriptome clusters onto their respective tissue sections revealed important information about the tumor microenvironment.

damage. These observations confirm our previous study which showed the presence of distinct immune and mutational profiles across the spectrum of angiosarcomas[5]. While individual biomarkers such as high TMB, elevated PD-L1 protein expression on IHC, and an immunogenic tumor microenvironment have been proposed to predict treatment response to immunotherapy, none of them have been able to robustly do so[9,11,12]. This perhaps suggest the potential of integrating various indicators, including mutational signatures, TMB levels and tumor inflammation signatures for this purpose[30–33].

Recently, spatial profiling technologies have provided valuable insights into understanding the tumor immune microenvironment and its implications on immune surveillance. In particular, the spatial organization of various cellular constituents enables the interpretation of tumor or immune-related signals in the context of the spatial architecture[34,35]. Indeed, in our exploratory analysis of AS-HN representative of each putative etiological subtype, we demonstrated that each of the tumor samples harbored distinct distributions of tumor cells and microenvironmental elements including fibroblasts and various immune cells. Importantly, clinically-relevant signals including the TIS score could be visualized and interpreted in the spatial context. We

showed that immune-hot angiosarcomas diagnosed from bulk tumor profiling may in fact carry most of the immune signals away from the tumor compartment. On the other hand, even immune-cold tumors may contain foci of inflammation within the tumor compartment. The clinical relevance of these observations will need further evaluation.

In conclusion, the current study described the genomic and immune landscape of Asian angiosarcomas of head and neck origin. Using spatial transcriptomics, we revealed the topological immune landscapes of AS-HN, with potential clinical implications to be validated in future studies.

## Methods

**Patient cohort.** Seventy-eight patients diagnosed with angiosarcoma at the Singapore General Hospital (SGH) and National Cancer Centre Singapore (NCCS) were identified. Snap frozen tissue samples were available from 23 patients, 18 of which with paired normal tissue or blood. An additional cohort of archival formalin-fixed paraffin-embedded (FFPE) angiosarcoma samples from 55 patients were available. Written informed consent for use of biospecimens and clinical data were obtained in accordance with the Declaration of Helsinki. This study has been approved by the Institutional Review Board of the National Cancer Centre Singapore (2010/426/B). Clinicopathological characteristics of all patients with angiosarcoma and the profiling methods applied to the study cohort are summarized in Supplementary Data 1.

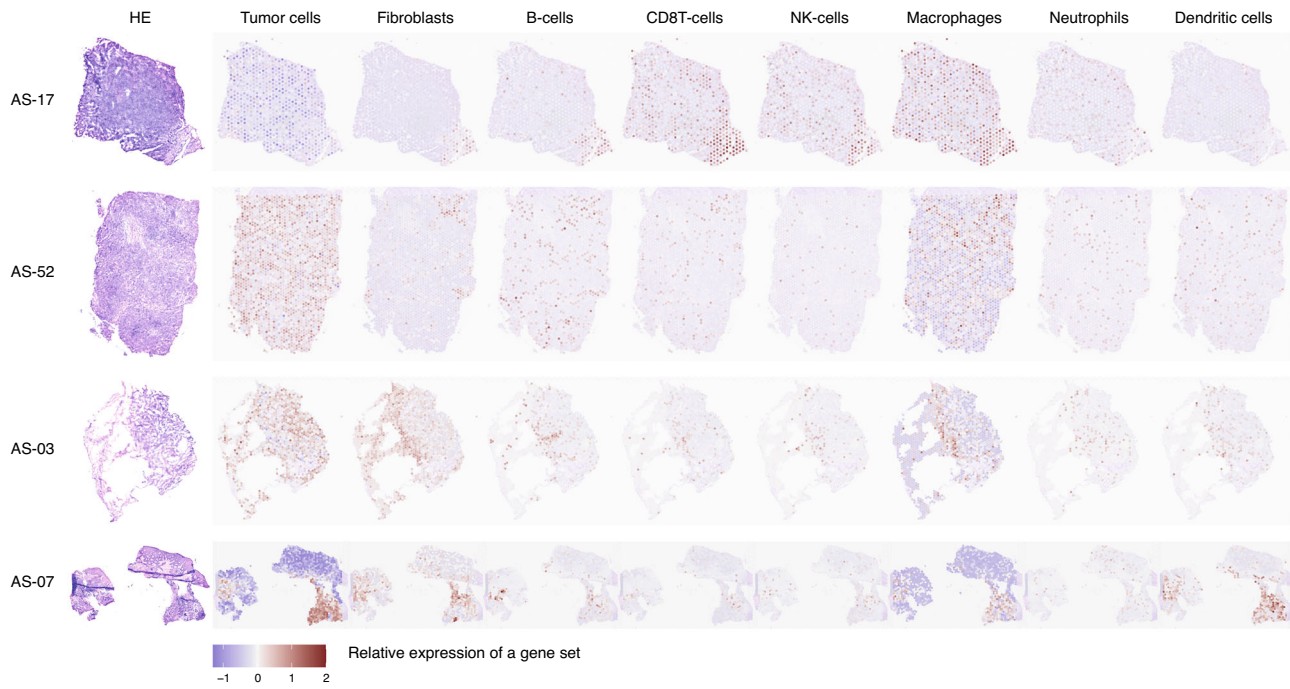

**Fig. 5 Spatial transcriptomics of cell populations in the tumor microenvironment.** H&E images of four head and neck angiosarcomas were shown, along with panels illustrating the estimated distribution of the tumor cells, stromal cells, as well as various immune cells in each sample.

**Cell lines.** Angiosarcoma cell lines (MO-LAS-B and ISO-HAS-B) were obtained from the Cell Resource Center for Biomedical Research, Institute of Development, Aging and Cancer (Tohoku University, Sendai, Japan), courtesy of Dr. Mikio Masuzawa (Kitasato University, Tokyo, Japan). MO-LAS-B was established from a patient with metastatic scalp lymphangiosarcoma to the pleura, whilst ISO-HAS-B was established from a patient with primary scalp haemangiosarcoma. Both cell lines were maintained in DMEM medium supplemented with 10% fetal bovine serum and 1% penicillin/streptomycin. Scalp angiosarcoma cell line AS-M was obtained from Johannes Gutenberg University, courtesy of Dr. James Kirkpatrick (Mainz, Germany) and maintained in endothelial cell growth medium (PromoCell, Heidelberg, Germany).

**Library preparation for whole genome sequencing (WGS).** DNA was extracted and purified with the FFPE RNA/DNA Purification Plus Kit (Norgen Biotek (Thorold, ON, Canada) according to manufacturer's instructions. A total of 76 angiosarcoma samples were subjected to whole genome sequencing, including snap frozen tumor samples ($n = 23$; 18 with paired normal), FFPE tumor samples ($n = 50$), and cell-lines ($n = 3$). This included 18 samples that were previously investigated[5]. Whole genome sequencing of snap frozen samples and cell-lines were performed on the Illumina HiSeq X platform as paired-end 150-base pair reads, using DNA inserts averaging 350 bp (NovogeneAIT Genomics Singapore Pte Ltd). For FFPE samples, we employed MGI DNBSEQ technology (DNA nanoball sequencing platform by MGI Tech, China) for WGS. All 50 libraries were sequenced, based on 100 bp paired-end reads, on the MGI DNBSEQ-T7 platform (MGI Tech, China).

**Somatic variant calling and generation of mutation signatures.** Read pairs were aligned to the human reference genome NCBI GRC Build 37 (hg19) using Burrows-Wheeler Aligner (BWA MEM, http://bio-bwa.sourceforge.net/)[36] and somatic mutations were identified by the Mutect2[37] variant caller with default parameters, following the standard GATK practices, including removal of PCR duplicates. Variants were subsequently annotated by Ensembl Variant Effect Predictor (VEP)[38]. TMB was estimated based on the proportion of nonsynonymous single nucleotide variants and short indels per coding megabase. Somatic mutational signatures were extracted using SigProfiler, an algorithm based on the 96 base substitution classification via non-negative matrix factorization, and compared with COSMIC v3 set of signatures[39].

**NanoString gene expression profiling.** We used the NanoString Pancancer IO360 panel (NanoString Technologies, Seattle, WA, USA) to interrogate gene expression, following manufacturer's protocol using the nCounter platform. Briefly, RNA was extracted from five 10 micron sections on all samples with adequate tumor tissue available and purified with the FFPE RNA/DNA Purification Plus Kit (Norgen Biotek (Thorold, ON, Canada) according to manufacturer's instructions, and analyzed using the 2100 Bioanalyzer (Agilent Technologies, Palo Alto, CA, USA). After excluding samples with suboptimal RNA integrity and content, the remaining samples were included in the nCounter analysis. Including 38 samples that were previously investigated[5], the final set of data passing QC

($n = 67$) were analyzed on the nSolver 4.0 Advanced Analysis module using default settings to derive differentially-expressed genes, pathway scores, and cell type scores. The panel was further analyzed using the Tumor Inflammation Signature (TIS) algorithm, which measures the level of immune infiltrate in a tumor and the tumor microenvironment[13]. This signature contains 18 genes related to antigen presentation, chemokine expression, cytotoxic activity, and adaptive immune resistance. A score is calculated as a weighted linear combination of the 18 genes' expression values normalized to stable housekeeper gene expression. High TIS was defined as more than or equal to the median score in the cohort.

**10X genomics visium platform.** Samples were embedded in TissueTek O.C.T compound (Sakura Finetek USA) and flash frozen in isopentane (Sigma-Aldrich; Merck), placed in a liquid nitrogen bath. RNA was extracted from four micron sections cut from the sample blocks chosen for the assay with the RNeasy Mini Kit (Qiagen). RNA quality was assessed by RNA integrity number (RIN) measured using TapeStation 4150 (Agilent Technologies). Haematoxylin (Dako) and eosin (Dako) staining (H&E) was performed on the sections to determine the morphology of the tissues. Optimization of the permeabilization timings for each sample was performed using Visium Tissue Optimization Slides and Reagent Kits (10X Genomics) with 10 µM sections. For the gene expression assay, 10 micron sections from the respective sample blocks were placed onto a Visium gene expression slide (10X Genomics). H&E staining was done according to the manufacturer's protocol. Permeabilization, reverse transcription, second strand synthesis and cDNA amplification was performed using the Visium Spatial Gene Expression Reagent Kit (10X Genomics). Dual indexed libraries were made with the Library Construction Kit (10X Genomics) and Dual Index Kit TT Set A (10X Genomics) according to the manufacturer's protocol. The final libraries were assessed using the Agilent Bioanalyzer High Sensitivity DNA kit and chip (Agilent Technologies). Loupe Browser 6.0 was used to estimate the capture area covered by the tissue within each frame on the slide to calculate the sequencing depths required. The libraries were pooled and sent for paired-end dual-indexed sequencing on the Novaseq 6000 instrument (Illumina).

**Analysis of spatial sequencing data.** The reads were demultiplexed and mapped against the hg38 reference genome using 10X Space Ranger v.1.3.1 (10X Genomics, CA, USA) with the default parameters for automatic alignment. Using Seurat v4.0, spatial data were first loaded into count matrices, and spots which had less than 10% of transcripts mapping to mitochondrial genes were retained for the scaling and normalization of genes expression measurements using sctransform approach, before merging (merge, Seurat) them to do an integrative analysis of dimensionality reduction and clustering. UMAP embeddings were used to visualize the spots distribution using 30 precomputed principal components in FindNeigbors and RunUMAP as well as with default resolution of 0.8 in FindClusters. The celltype(s) of each spot was established with PanglaoDB Augmented 2021 database[40], in addition to the gene expression profile of some cell type-specific markers (as

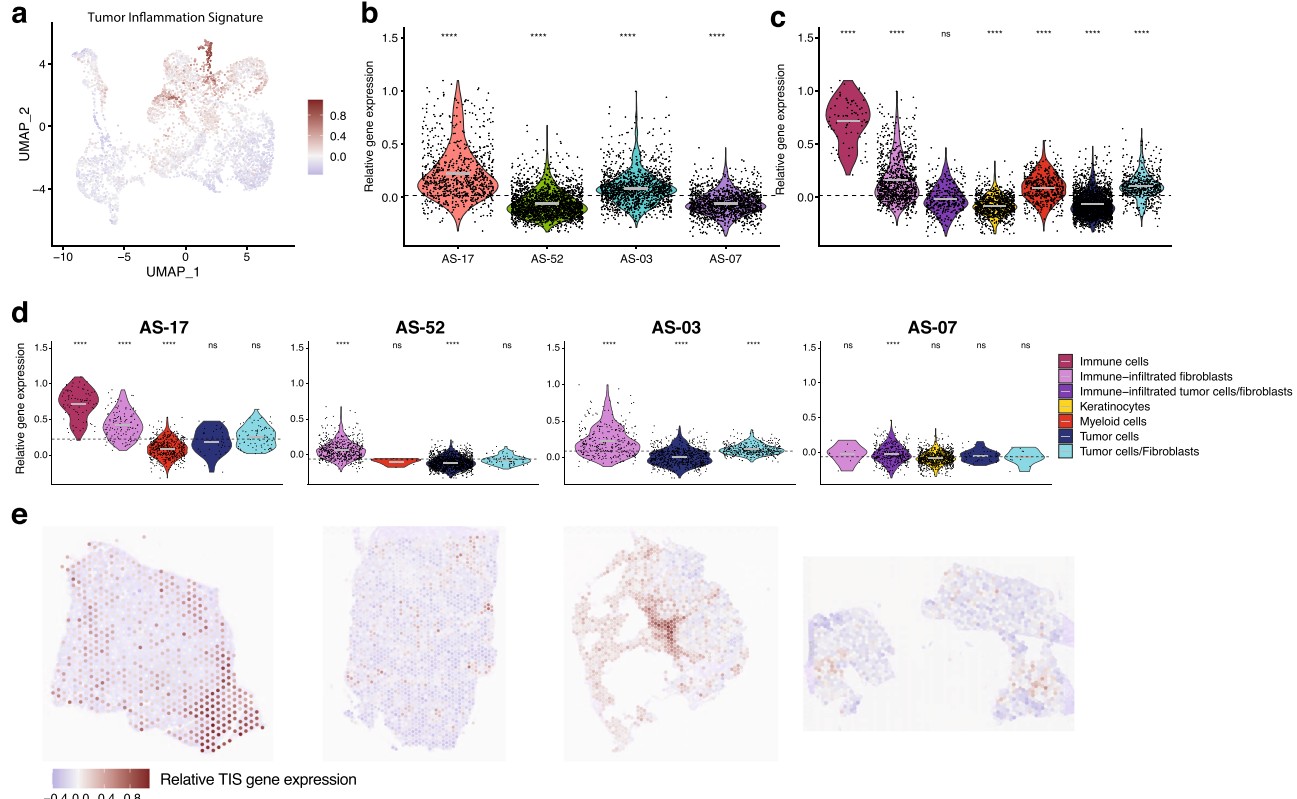

**Fig. 6 Spatial projections of the Tumor Inflammation Signature (TIS) in head and neck angiosarcoma. a** UMAP plot of spot transcriptome clusters showing relative average expression of genes informing the TIS score. Violin plots showing relative average expression of TIS genes across (**b**) the four samples and (**c**) the clusters annotated by their predominant cell types. **d** Immune cells from AS-17 exhibited the highest average expression of TIS genes, followed by immune-infiltrated fibroblasts in AS-17, AS-03 and AS-52, as well as immune-infiltrated tumor cells/fibroblasts in AS-07. The myeloid cells had the lowest TIS gene expression in AS-17. The dotted horizontal line represents the mean TIS genes expression of a sample, while the gray line in each violin represents the mean TIS genes expression of a cell type in a sample. ****, global *p*-value < 0.0001 by the Wilcoxon test. **e** Relative average TIS gene expression level was visualized with respect to their spatial context in each sample. TIS signals coincided with the immune compartment but were largely tumor-excluded in AS-17. The corresponding spots exhibiting higher levels of TIS genes expression manifested a focal pattern in AS-03 and AS-07, while displaying a scattered pattern in AS-52.

indicated in Supplementary Fig. 3). Cell types with less than three spots, in each sample, were omitted from the downstream analysis. No batch-effect correction was performed to maintain the clustering architecture when overlaying on the tissue morphology of each sample.

Eighteen genes (*PSMB10, CD276, PDCD1LG2, HLA-DQA1, NKG7, HLA-E, TIGIT, CD274, CXCL9, CMKLR1, LAG3, CD26, CXCR6, IDO1, CD8A, HLA-DRB1, CCL5,* and *STAT1*) from the NanoStringIO360 panel were used to identify TIS signatures (AddModuleScore, Seurat) in characterizing immune hot and cold specimens. In addition, selected genes from the same panel (tumor cells: *PECAM1* and *ERG*; fibroblasts: *FBLN1, FAP,* and *DES*; B-cells: *CD79A* and *CD79B*; CD8T-cells: *CD8A* and *CD8B*; NK-cells: *KLRK1* and *KLRD1*; macrophages: *CD14* and *CD68*; neutrophils: *CSF3R* and *FPR1*, and Dendritic cells: *CD209* and *CCL13*) were used to annotate (AddModuleScore, Seurat) immune cells. Wilcoxon test, with Holm-Bonferroni correction, was conducted to statistically evaluate the differential expression of TIS genes in each cell type against the base mean of each sample.

**Statistics and reproducibility**. Comparisons of the frequencies of categorical variables were performed using Pearson's Chi-squared tests or Fisher's exact tests. Continuous variables were represented by Box–Whisker plots and their associations with categorical variables were evaluated by Mann–Whitney U tests. All statistical analyses were conducted assuming a two-sided test with significance level of 0.05 unless otherwise stated, and performed using MedCalc for Windows, version 18.2.1 (MedCalc Software, Ostend, Belgium).

**Reporting summary**. Further information on research design is available in the Nature Portfolio Reporting Summary linked to this article.

## Data availability

Spatial transcriptomic data and WGS data were deposited in the European Genome-phenome Archive (EGA) under accession no. EGAD00001005366 and EGAD00001010140, as well as Gene Expression Omnibus (GEO) under accession no. GSE227469. The NanoString gene expression profiling data generated during the current study are available in GEO under accession no. GSE226338. Source data underlying main figures are provided in Supplementary Data 1-2 (Figs. 1 and 2), Supplementary Data 3-4 (Fig. 3b-f) and Supplementary Data 5 (Fig. 6b, d).

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

## Acknowledgements

This work was supported by the Singapore Ministry of Health's National Medical Research Council under its Singapore Translational Research Investigator Award (NMRC/STAR/0006/2009), Transition Award (TA21jun-0005), RTF Seed Fund (SEEDFD21jun-0002), and TETRAD II collaborative centre grant (CG21APR2002), as well as SingHealth Duke-NUS Oncology Academic Clinical Programme (08/FY2020/EX/75-A151). Whole genome sequencing was supported by MGI Tech. We would like to thank all subjects who have participated in this study.

## Author contributions

J.W.L. analyzed the data and drafted the manuscript; J.W.L., A.H.L, P.G., J.Q.L. performed the bioinformatics analyses; J.Y. provided pathological assessment of tissues; J.Y.C provided patient samples and clinical data; B.Y.L, B.K., E.C.Y.L, J.Y.L, N.X.G., T.K.K., J.C.T.L., C.C.N. and C.K.O. provided technical expertise and performed various experiments. J.Y.C. and T.B.T. conceived the study, interpreted the results, and revised the manuscript; and all authors read and approved the final version of the manuscript.

## Competing interests

J.Y.C. received research support from MGI Tech. N.X.G. is an employee of MGI Tech. The other authors declare no competing interests.
