## [Peer Review File · Communications Biology]

Reviewers' comments:

Reviewer #1 (Remarks to the Author):

Jui Wan LOH and colleagues investigated the topological immune landscapes of Asian head and neck angiosarcoma, integrating multi-omics data. Overall, this study could provide some insights for AS-HN immunology and might have some potential to improve implementation of treatment strategies. But the manuscript has some vagueness in the methods section and lacks sufficient reference citations. In general, I would like to make the following suggestions for improvement.

1. Parameters for Space Ranger and Seurat need to be specified.
2. The reference for PanglaoDB needs to be cited.
3. The authors should explain clearly how each type was annotated and specify markers used for each cell type.
4. What if a cluster express canonical markers for multiple cell types. What evidences were used to annotate and validate cell identity.
5. In the Data and materials availability section, the author should upload both raw data and corresponding matrices to public databases (NCBI/EBI) and provide accession ID for ST, WGS and NanoString data.

Reviewer #2 (Remarks to the Author):

In this study, LOH et al examined 81 cases of angiosarcomas by integrating information from multi-omic information. they observed biases of tumor mutation burden (TMB) between UV positive vs negative. NanoString profiling identified three clusters with distinct tumor inflammation signature (TIS) scores. By spatial transcriptomics, they identified dominant but tumor-excluded inflammatory signals in "immune-hot" cases and immune foci even in otherwise "cold" cases.

This is generally an interesting manuscript. However, some biological aspects (somatic mutation and NanoString profiling of angiosarcomas) of this manuscript have been published before (PMID: 33016928). What indeed is new is the data of spatial transcriptomics. From my point of view, this is the most interesting aspect of the study. However, I would like additional evidence for the Nanostring profiling and spatial transcriptomics data. For example, they identified that around 1/3 of samples displayed immune hot clustered. This conclusion is drawn by transcriptome and additional data to prove the presence of infiltrated immune cells by confocal should be provided to convince the conclusion. They also detected an immune-hot case by spatial transcriptomics with more than 85% of myeloid and immune cells. However, I am not convinced by the data. Since there are so many immune cells in the specimen, which is uncommon in tumor tissue. I am not sure if it's a representative section of tumor tissue. This may happen if it is close to the vessel or inflammatory sites.

Appendix I

COMMSBIO-23-0177

Reviewer #1 (Remarks to the Author):

Jui Wan LOH and colleagues investigated the topological immune landscapes of Asian head and neck angiosarcoma, integrating multi-omics data. Overall, this study could provide some insights for AS-HN immunology and might have some potential to improve implementation of treatment strategies. But the manuscript has some vagueness in the methods section and lacks sufficient reference citations. In general, I would like to make the following suggestions for improvement.

AUTHOR REPLY: We thank the Reviewer for the positive comments. We have made several changes to improve the *Methods* section, and also cited the relevant references in the revised manuscript as detailed in the point-by-point responses below:

1. Parameters for Space Ranger and Seurat need to be specified.

AUTHOR REPLY: We apologize for the lack of specifications in the earlier manuscript. We have now added the parameters and description for running Space Ranger, as well as for Seurat. The *Methods* section has also been generally improved with the following: “*The reads were demultiplexed and mapped against the hg38 reference genome using 10X Space Ranger v.1.3.1 (10X Genomics, CA, USA) with the default parameters for automatic alignment. Using Seurat v4.0, spatial data were first loaded into count matrices, and spots which had less than 10% of transcripts mapping to mitochondrial genes were retained for the scaling and normalization of genes expression measurements using sctransform approach, before merging (merge, Seurat) them to do an integrative analysis of dimensionality reduction and clustering. UMAP embeddings were used to visualize the spots distribution using 30 precomputed principal components in FindNeighbors and RunUMAP as well as with default resolution of 0.8 in FindClusters. The celltype(s) of each spot was established with PanglaoDB Augmented 2021 database (Franzen et al., 2019), in addition to the gene expression profile of some cell type-specific markers (as indicated in Supplementary Figure S3). Cell types with less than three spots, in each sample, were omitted from the downstream analysis. No batch-effect correction was performed to maintain the clustering architecture when overlaying on the tissue morphology of each sample.*”

The *Reference* section has been updated accordingly: Franzén O, Gan LM, Björkegren JLM. PanglaoDB: a web server for exploration of mouse and human single-cell RNA sequencing data. Database (Oxford). 2019 Jan 1;2019:baz046. doi: 10.1093/database/baz046. PMID: 30951143; PMCID: PMC6450036.

2. The reference for PanglaoDB needs to be cited.

AUTHOR REPLY: We have included the citation for PanglaoDB in the revised manuscript (Franzén et al. 2019). Changes to manuscript are summarized in *Response 1* above.

The *Reference* section has been updated accordingly: Franzén O, Gan LM, Björkegren JLM. PanglaoDB: a web server for exploration of mouse and human single-cell RNA sequencing data. Database (Oxford). 2019 Jan 1;2019:baz046. doi: 10.1093/database/baz046. PMID: 30951143; PMCID: PMC6450036.

3. The authors should explain clearly how each type was annotated and specify markers used for each cell type.

AUTHOR REPLY: We are grateful to the Reviewer for highlighting this point. In the revised manuscript, under *Methods*, we have added the phrase “as indicated in Supplementary Figure S3”. Figure S3 shows the markers used for each cell type annotation in Figure 4. As for Figure 5, the selected genes from the NanoString IO360 panel are now included in the *Methods* section. We attempted to annotate the cell types with our best effort in identifying the cell-type specific markers that are up-regulated in each group. Changes to the manuscript are summarized in *Response 1* above.

In addition, the following has also been included in the *Methods* section: “Eighteen genes (*PSMB10*, *CD276*, *PDCD1LG2*, *HLA-DQA1*, *NKG7*, *HLA-E*, *TIGIT*, *CD274*, *CXCL9*, *CMKLR1*, *LAG3*, *CD26*, *CXCR6*, *IDO1*, *CD8A*, *HLA-DRB1*, *CCL5*, and *STAT1*) from the NanoStringIO360 panel were used to identify TIS signatures (*AddModuleScore*, *Seurat*) in characterizing immune hot and cold specimens. In addition, selected genes from the same panel (tumor cells: *PECAM1* and *ERG*; fibroblasts: *FBLN1*, *FAP*, and *DES*; B-cells: *CD79A* and *CD79B*; CD8T-cells: *CD8A* and *CD8B*; NK-cells: *KLRK1* and *KLRD1*; macrophages: *CD14* and *CD68*; neutrophils: *CSF3R* and *FPR1*, and Dendritic cells: *CD209* and *CCL13*) were used to annotate (*AddModuleScore*, *Seurat*) immune cells. Wilcoxon test, with Holm-Bonferroni correction, was conducted to statistically evaluate the differential expression of TIS genes in each cell type against the base mean of each sample.”

4. What if a cluster express canonical markers for multiple cell types. What evidences were used to annotate and validate cell identity.

AUTHOR REPLY: In the case of high expression from multiple cell-type specific markers, we labelled them as the mixture of cell types (eg. immune-infiltrated tumor cells/fibroblasts), which occurs most of the time, since each spot on the 10X Visium slide consists of approximately 5-10 cells, resulting in a possibility for different cell types to be captured in the same spot. Changes to manuscript are summarized in *Response 1* above.

5. In the Data and materials availability section, the author should upload both raw data and corresponding matrices to public databases (NCBI/EBI) and provide accession ID for ST, WGS and NanoString data.

AUTHOR REPLY: We have uploaded the raw data and corresponding metadata to public databases. Spatial transcriptomic data and WGS data were deposited in the European Genome-phenome Archive (EGA) under accession no. EGAD00001005366 and EGAD00001010140, as well as Gene Expression Omnibus (GEO) under accession no. GSE227469. The NanoString gene expression profiling data generated during the current study are available in GEO under accession no. GSE226338. The above information has been included in the revised manuscript under *Data and materials availability* as follows “*Spatial transcriptomic data and WGS data were deposited in the European Genome-phenome Archive (EGA) under accession no. EGAD00001005366 and EGAD00001010140, as well as Gene Expression Omnibus (GEO) under accession no. GSE227469. The NanoString gene expression profiling data generated during the current study are available in GEO under accession no. GSE226338.*”

Reviewer #2 (Remarks to the Author):

In this study, LOH et al examined 81 cases of angiosarcomas by integrating information from multi-omic information. They observed biases of tumor mutation burden (TMB) between UV positive vs negative. NanoString profiling identified three clusters with distinct tumor inflammation signature (TIS) scores. By spatial transcriptomics, they identified dominant but tumor-excluded inflammatory signals in “immune-hot” cases and immune foci even in otherwise “cold” cases.

This is generally an interesting manuscript. However, some biological aspects (somatic mutation and NanoString profiling of angiosarcomas) of this manuscript have been published before (PMID: 33016928). What indeed is new is the data of spatial transcriptomics. From my point of view, this is the most interesting aspect of the study.

However, I would like additional evidence for the Nanostring profiling and spatial transcriptomics data. For example, they identified that around 1/3 of samples displayed immune hot clustered. This conclusion is drawn by transcriptome and additional data to prove the presence of infiltrated immune cells by confocal should be provided to convince the conclusion. They also detected an immune-hot case by spatial transcriptomics with more than 85% of myeloid and immune cells. However, I am not convinced by the data. Since there are so many immune cells in the specimen, which is uncommon in tumor tissue. I am not sure if it's a representative section of tumor tissue. This may happen if it is close to the vessel or inflammatory sites.

AUTHOR REPLY: We thank the Reviewer for the positive comments. The NanoString IO 360 Immune clusters were derived from the 18-gene Tumor Inflammation Signature (TIS) that measures a peripherally-suppressed, adaptive immune response and has been shown to correlate with response to checkpoint inhibitors. Individual immune cell types present within the tumor likely do not fully represent the TIS score in its entirety. Nonetheless, we examined the samples profiled with NanoString and compared with available data on immune cell infiltration from our prior publication (PMID: 33016928), obtained using multiplex immunohistochemistry/immunofluorescence (mIHC/IF). We observed that the proportion of immune cells (CD8+ cytotoxic T-cells, CD15+ neutrophils, CD68+ macrophages, FOXP3+ T-reg cells) relative to ERG+ tumor cells correlated with the immune clusters (n = 11 “hot” and n = 5 “cold” cases) inferred from NanoString transcriptomic profiling ($p = 0.020111$) (see *Response Figure 1*). This information has now been included in the revised manuscript in the *Results* section and *Response Figure 1* has been incorporated into Figure S2b).

In the immune-hot case by spatial transcriptomics, it could be that the myeloid and immune cell numbers are overestimated since each spot on the 10X Visium slide consists of approximately 5-10 cells. Nonetheless, in the immune-hot case, the signals of these immune cells appeared to be distributed widely across 86% of the tissue area, which also contains tumor cells and fibroblasts. To confirm the tumor content, we had performed mIHC/IF on this case and showed that tumor cells (ERG+) account for 60.3% of the total cellular content (*Response Figure 2*). These observations are highlighted in the revised manuscript under the *Results* section, *Spatial analysis of head and neck angiosarcoma*, as follows: “Collectively, they make up the major composition (86%) of the tissue area, along with fibroblasts and tumor cells.”

Response Figure 1. The proportion of immune cells (CD8+ cytotoxic T-cells, CD15+ neutrophils, CD68+ macrophages, FOXP3+ T-reg cells) relative to ERG+ tumor cells were correlated with the NanoString immune clusters (Mann-Whitney U test, $p = 0.020211$).

Response Figure 2. Representative mIHC/IF images of various immune cells (CD8+ cytotoxic T-cells, CD15+ neutrophils, CD68+ macrophages, FOXP3+ T-reg cells) and tumor cells (ERG+) in sample AS-17.

REVIEWERS' COMMENTS:

Reviewer #1 (Remarks to the Author):

All the mentioned questions and concerns have been clearly explained.

Reviewer #2 (Remarks to the Author):

My questions are addressed and I have no further comments.